# Poliovirus-Neutralizing Antibody Seroprevalence and Vaccine Habits in a Vaccine-Derived Poliovirus Outbreak Region in the Democratic Republic of Congo in 2018: The Impact on the Global Eradication Initiative

**DOI:** 10.3390/vaccines12030246

**Published:** 2024-02-27

**Authors:** Megan Halbrook, Adva Gadoth, Patrick Mukadi, Nicole A. Hoff, Kamy Musene, Camille Dzogang, Cyrus Shannon Sinai, D’Andre Spencer, Guillaume Ngoie-Mwamba, Sylvia Tangney, Frank Salet, Michel Nyembwe, Michel Kambamba Nzaji, Merly Tambu, Placide Mbala, Trevon Fuller, Sue K. Gerber, Didine Kaba, Jean Jacques Muyembe-Tamfum, Anne W. Rimoin

**Affiliations:** 1Department of Epidemiology, Jonathan and Karin Fielding School of Public Health, University of California, Los Angeles, CA 90095, USA; advag@ucla.edu (A.G.); nhoff84@ucla.edu (N.A.H.);; 2National Institute of Biomedical Research, Kinshasa P.O. Box 1197, Democratic Republic of the Congo; patrickmukadi@gmail.com (P.M.); dzogangcamille@gmail.com (C.D.);; 3Carolina Population Center, University of North Carolina, Chapel Hill, NC 27516, USA; sinai@unc.edu; 4Children’s National Research Institute, Center for Genetic Medicine Research, 111 Michigan Avenue NW, Washington, DC 20010, USA; dandrespencer@g.ucla.edu; 5Expanded Programme for Immunization, Kinshasa P.O. Box 1197, Democratic Republic of the Congo; 6Bill and Melinda Gates Foundation, Seattle, WA 98109, USA; 7Institute of the Environment and Sustainability, University of California, Los Angeles, CA 90095, USA; 8Kinshasa School of Public Health, University of Kinshasa, Kinshasa P.O. Box 11850, Democratic Republic of the Congo

**Keywords:** poliovirus, serosurvey, Democratic Republic of the Congo, vaccine coverage, cVDPV, SIAs, OPV, IPV, nOPV2

## Abstract

Despite the successes in wild-type polio eradication, poor vaccine coverage in the DRC has led to the occurrence of circulating vaccine-derived poliovirus outbreaks. This cross-sectional population-based survey provides an update to previous poliovirus-neutralizing antibody seroprevalence studies in the DRC and quantifies risk factors for under-immunization and parental knowledge that guide vaccine decision making. Among the 964 children between 6 and 35 months in our survey, 43.8% (95% CI: 40.6–47.0%), 41.1% (38.0–44.2%), and 38.0% (34.9–41.0%) had protective neutralizing titers to polio types 1, 2, and 3, respectively. We found that 60.7% of parents reported knowing about polio, yet 25.6% reported knowing how it spreads. Our data supported the conclusion that polio outreach efforts were successfully connecting with communities—79.4% of participants had someone come to their home with information about polio, and 88.5% had heard of a polio vaccination campaign. Additionally, the odds of seroreactivity to only serotype 2 were far greater in health zones that had a history of supplementary immunization activities (SIAs) compared to health zones that did not. While SIAs may be reaching under-vaccinated communities as a whole, these results are a continuation of the downward trend of seroprevalence rates in this region.

## 1. Background

Significant progress has been made toward polio eradication. Of the three polio serotypes, currently, only wild-type poliovirus (WPV) type 1 is actively circulating in two remaining countries: Afghanistan and Pakistan [1]. Globally, the last case of WPV2 was observed in 1999 and was officially declared eradicated by the World Health Organization (WHO) in 2015. The last case of WPV3 was observed in Nigeria in 2012, and on 24 October 2019, the WHO declared it eradicated [2]. In 2012, the WHO General Assembly released a strategic plan for polio eradication that called for the eventual removal of the oral polio vaccine (OPV). In April 2016, a switch day was coordinated globally where trivalent OPV (tOPV) containing poliovirus types 1, 2, and 3 were replaced with bivalent vaccines containing polio types 1 and 3 only. In the Democratic Republic of the Congo (DRC), importations of WPV1 and WPV3 from India by way of neighboring Angola caused sustained outbreaks from 2006 to 2011 [3,4]. In DRC, the last case of WPV3 was confirmed on 24 June 2009, and the last case of WPV1 was reported in Maniema province, with an onset date of 20 December 2011 [5,6,7].

Despite the successes in WPV surveillance and control, poor vaccine coverage in many areas of the DRC has led to the occurrence of circulating vaccine-derived poliovirus (cVDPV) outbreaks [8,9]. Cases of cVDPVs can occur in under-immunized populations when Sabin-strain poliovirus is excreted following immunization and transmitted to unprotected populations, which can cause acute flaccid paralysis (AFP); with continued circulation and mutation, these strains can eventually revert to wild-type virus [10,11]. In 2011–2012, there was a large outbreak of 30 cVDPV cases in Haut Lomami province in southeastern DRC. Since then, there has been a concentration of cases in this region— the former Katanga province, which is now divided into four new provinces as part of DRC’s newly decentralized provincial jurisdictions: Lualaba, Haut-Katanga, Haut Lomami, and Tanganyika provinces. This concentration of cVDPV2 cases raises questions about the vaccination landscape of this region. In the wake of the global cessation of the trivalent oral polio vaccine (tOPV) and the resulting immunity gap of polio type 2 among new birth cohorts, areas with low vaccine coverage rates were at increased risk for cVDPV2 outbreaks. Indeed, in February 2017, cVDPV2 reemerged—a two-case cluster was observed in Maniema province, and later that year, a twenty-seven-case cluster originating in Haut Lomami and spanning three other neighboring provinces was also observed [12]. In 2018, 20 cases of cVDPV were reported from five provinces. Of these, 11 cVDPV2 cases were reported from Mongala province in the northwest of the country. In the southeast, Haut Lomami reported two cases of cVDPV2, Tanganyika reported three cases of cVDPV2, and Haut Katanga province reported four cases of cVPDV2. Additionally, the dramatic increase in case reports from Mongala province in 2018 amplifies a broader nationwide concern regarding low poliovirus vaccination rates. On 13 February 2018, the DRC Ministry of Health declared cVDPV2 to be a national public health emergency [13]. To combat cVDPV emergence, it is the vaccination policy of the DRC Ministry of Health and the WHO Expanded Programme on Immunization (EPI) to implement supplementary immunization campaigns (SIAs) in regions where cVDPV2 risk is high or cases have recently been observed. In addition to the four-dose vaccine series recommended as a part of routine immunization—the bivalent oral polio vaccine (bOPV; introduced April 2016) and the inactivated polio vaccine (IPV; introduced in 2015)—SIAs are mass door-to-door vaccination campaigns that offer additional OPV doses to all children under the age of five present in a community, regardless of prior vaccination status. In the DRC, monovalent oral polio vaccine type 2 (mOPV2) or bOPV is used during SIA campaigns. This study provides an update to previous poliovirus seroprevalence studies that have occurred in the DRC (2014) [6] and in the former Katanga province (2016) [14] and seeks to understand risk factors for under-immunization and parental knowledge and behaviors that guide vaccine decision making. 

## 2. Methods

### 2.1. Study Sample

This study was designed as a cross-sectional population-based survey modeled after the USAID Demographic and Health Survey, with the intention of providing an update on population seroprevalence of markers of poliovirus immunity in an outbreak-prone region of southeastern DRC. Field research was conducted in March 2018 in eight health zones across Haut Lomami (HL) and Tanganyika (T) provinces (HL: Butumba, Lwamba, Malemba-Nkulu, Mukanga; T: Ankoro, Manono, Kabalo, Kongolo). Health zones were chosen and grouped based on the number of cVDPV2 cases and SIAs performed prior to study commencement such that study enrollment was conducted in health zones at different levels of risk for cVDPV2 emergence and different numbers of interventions in the 12 months prior. Four health zones in Haut Lomami province had a history of cV2DPV cases and had 4–5 SIAs in the past year, Ankoro and Manono health zones in Tanganyika province experienced cVDPV2 cases and had 2 SIAs in the past year, and Kabalo and Kongolo health zones in Tanganyika did not have any reported cVDPV cases or SIAs (Figure 1). 

Within health zones, five study sites were selected via stratified random sampling in identified health districts using satellite imagery-derived settlement feature layers. Villages were randomly selected using ArcGIS 10.8.2 software’s Create Random Point tool using two parameters: selected settlements did not fall in the same administrative health area and had a minimum separating distance of 500 m. All houses in each selected village were sampled until the necessary sample size was met. Households that refused to participate in this study were marked as refusals in the tablet-based questionnaire (2.14%). This selection method was used to reduce the bias extending from use of microplans, census-like documents used by the EPI, which can unintentionally exclude individual villages or clump multiple villages together. At each eligible household, all healthy children between 6 and 35 months and their parents or guardians were invited to participate. Study requirements consisted of a questionnaire administered by trained study staff and blood samples collected via dried blood spot (DBS). If available, vaccine cards were photographed. Prior to enrollment, community leaders were visited at each study site to educate, sensitize, and inform community members about vaccinations and vaccine-preventable diseases. Informed consent was administered orally in French or Swahili by study administrators.

### 2.2. Laboratory Analysis

All collected samples were initially processed at DRC’s National Institute of Biomedical Research (INRB) in Kinshasa, with one dried blood spot per child shipped to the US Centers for Disease Control and Prevention (CDC) in Atlanta for polio testing. The methods used for laboratory analysis of serologic samples have been previously described [15]. Briefly, sera and extracted DBS were processed using the polio microneutralization assay, and neutralization titers were reported in a log_2_ format, with 2.5 log_2_ as the lower limit of detection and 10.5 log_2_ as the upper limit of detection. Neutralizing antibodies were assessed against poliovirus serotypes 1, 2, and 3, and titers ≥ 3.0 log_2_ were considered evidence of seroprotection.

### 2.3. Statistical Analysis

Frequencies, chi-square tests of proportions, and logistic regression models were performed to quantify the relationship between various demographic, knowledge, and behavior variables (Table 1) and population seroprevalence to each of three poliovirus subtypes. While complete polio vaccination requires immunity to all three subtypes, many participants have antibodies to none, some, or all polio serotypes. Owing to the fact that three distinct poliovirus serotypes can cause polio disease and OPV can contain differing combinations of serotype protections, six polio seroprofiles—none; any reactivity to types 1, 2, and 3; seroreactivity to type 2 only; and seroreactivity to all three types—were used as the analytical framework for polio immunity. Analyses were performed using SAS version 9.6 (SAS Institute, Cary, NC, USA), maps were generated using ArcGIS software version 10.5 (ESRI, Redlands, CA, USA), and figures were generated using the ggplot2 package for R (R Core Team, 2014). Ethical approval for this study was obtained via University of California, Los Angeles’ Institutional Review Broad (IRB#18–000303) and the Ethics Committee at the Kinshasa School of Public Health, University of Kinshasa, in the DRC (ESP-CE-027–2018).

## 3. Results

Among the 964 participants in our survey, 43.8% (95% CI: 40.6–47.0%), 41.1% (38.0–44.2%), and 38.0% (34.9–41.0%) had protective neutralizing titers to polio types 1, 2, and 3, respectively. Mean neutralizing antibody titers for each serotype were found to be statistically different via ANOVA test (*p* < 0.0001): 4.73 IU/mL, 3.75 IU/mL, and 4.08 IU/mL for polio types 1, 2, and 3, respectively. Seroprevalence varied between individuals: 17.9% (n = 172) had neutralizing antibodies for all three polio serotypes, 36.4% (n = 351) had none, and 45.7% (n = 441) had varying combinations of poliovirus serotypes (Figure 2).

In three health zones (Butumba, Malemba-Nkulu, and Mukanga), which had been previously sampled in a 2016 survey of the region, polio seroprevalence fell an average of 32.6% (range: 16.6–52.0%) (Figure 3). Poliovirus-neutralizing antibody seroprevalence rates increased with age for all serologic profiles except among those who had antibodies to type 2 only. Seroprevalence rates across polio serotypes ranged from 29.8 to 34.6% for 6–11-month-olds, 41.7 to 48.3% for 12–23-month-olds, and 47.4 to 50.0% for those 24 months and older. Across each age group, the lowest seroprevalence was displayed for poliovirus type 3 (Appendix A). Notably, a number of children in the home and travel time to health facilities were not associated with polio vaccine seroprevalence.

We also assessed parental knowledge of polio and found that 60.7% of parents and guardians in this study reported knowing what polio was, yet only 25.6% reported knowing how it spreads. When asked about the symptoms of polio, 74.4% of respondents correctly identified paralysis, 62.6% identified fever, and 32.6% identified diarrhea; 14.7% reported that they did not know the symptoms of poliovirus. Our data found that polio outreach efforts were successfully connecting with communities—79.4% of participants reported that a community health worker had come to their home with information about polio, and 88.5% had heard of a polio vaccination campaign (Table 1).

Proportions of seroprevalence were similar between the Haut Lomami health zones (n = 4) and Ankoro and Manono—the six health zones that experienced cVPDV2 cases and SIAs. These health zones had higher seroprevalence rates for serotype 2 and participants with all three antibodies and lower rates of no antibodies compared with Kabalo and Kongolo, health zones that had no cVDPV2 cases or SIAs (*p* < 0.0001). In the Kabalo and Kongolo health zones, 46.1% of the sample population had no poliovirus-neutralizing antibodies. In a logistic regression model, estimating seroprevalence of each seroprofile predicted by health zone SIA history, controlling for child age, parental knowledge of polio, and having a recent home visit, age and health zone history were significant predictors of seroreactivity (Figure 4). Increasing age was positively associated with the odds of seroprevalence, with the exception of those with only markers of type 2 antibodies, as described above. The odds of seroreactivity to poliovirus type 2, only type 2, and to all serotypes were increased for health zones that had cVDPV2 cases and SIA campaigns compared to the two health zones that did not. The odds of seroreactivity to type 2 in Haut Lomami health zones, which had 4–5 SIAs, was 4.41 (95% CI: 3.07–6.35) times that of the health zones with none. The odds of seroreactivity to type 2 in Ankoro and Manono health zones, which had two SIAs, was 4.60 (95% CI: 3.17–6.69) times that of the health zones with no SIA activities, controlling for other factors. The odds of seroreactivity to only serotype 2 were far greater in health zones that had a history of SIAs within the last year compared to health zones that did not; Haut Lomami adjusted odds ratio (aOR) was 6.03 (95% CI: 2.89–12.6); Ankoro and Manono aOR was 5.31 (95% CI: 2.52–11.21).

Seroprevalence of poliovirus type 1 and type 3 was not associated with the recent SIA history of a health zone. As the number of SIAs increased in a health zone, so did the likelihood of both having someone visit your home to distribute information about polio and hearing about a polio vaccination campaign. The odds of having a home visit in Ankoro and Manono, health zones that both experienced two SIAs, were 1.06 (95% CI: 0.736–1.53) times that of health zones that had no SIA activity; in Haut Lomami, which experienced four to five SIAs, the odds were 1.90 (95% CI: 1.27–2.84) times greater than control health zones. Similarly, the odds of hearing about a mass immunization campaign increased with the number of campaigns performed in the respondent’s residential health zone. In Ankoro and Manono (two SIAs), the odds of hearing of a campaign were 2.25 (95% CI 1.41–3.59) times greater than in control health zones with no SIAs. In the four health zones of Haut Lomami (four to five SIAs), the odds of having heard about a campaign were 3.59 (95% CI: 2.10–6.13) times greater.

## 4. Discussion

These results are a continuation of the downward trend of seroprevalence rates in this region. In 2014, a national serologic survey observed seroprevalence rates in the former Katanga province (which includes the current Haut Lomami and Tanganyika provinces) of 75–80% for type 1, 85–90% for type 2, and 70–75% for type 3 [6]. In 2016, we conducted a cross-sectional serosurvey of the same design in eight health zones in Haut Lomami province (including Butumba, Malemba-Nkulu, and Mukanga surveyed again in this study). That survey found the overall seroprevalence rates to be 79.8% (95% CI: 77.7–81.8%) for type 1, 91.7% (CI: 90.3–93.1%) for type 2, and 70.5% (CI: 68.2–72.8%) for type 3 [14]. Since then, overall estimated seropositivity rates in this region have fallen to 43.8% (CI: 40.6–47.0%), 41.1% (38.0–44.2%), and 38% (34.9–41.0%) for polio types 1, 2, and 3, respectively. Similar to our 2016 findings, age was a major predictor of seroprevalence, as the older a child is, the greater the opportunity for routine vaccination or involvement in an SIA. We also found that only 17.9% of children surveyed had antibodies to all three poliovirus serotypes and many had a diverse mix of serologic immunity profiles, likely a reflection of highly variable and inconsistent polio vaccine distribution, availability, and uptake in the region. 

During routine vaccination with bOPV, type 1 and type 3 are always coupled; thus, it was expected that seroprevalence rates of these two serotypes would be roughly equal. However, we saw a significantly greater percentage of children with seroreactivity to type 1 than to type 3. This result is most likely a result of the reduced immunogenicity of serotype 3 in the Sabin vaccine; on average, the observed neutralizing antibody titers to serotype 1 were greater than that of serotype 3 [16]. Moreover, seroreactivity to only type 2 was observed in 10.8% of our study participants (n = 104). This was the only serologic profile that was not associated with age, likely indicating that this group was either vaccinated for the first time during a supplementary vaccination campaign via mOPV2 or was infected with cVDPV2. Overall, the patchwork of vaccination campaigns and the use of multiple different OPV vaccines (tOPV, bOPV, and mOPV2) makes tracking and quantifying the overall successes and failures of the cVPDV response in the region very difficult. Vaccination cards are often used to track received vaccinations, but in this rural cohort, just 13.1% had a vaccination card. The global polio eradication strategy, which informs national policies, only requires routine immunizations are captured on vaccination cards, leaving SIA vaccines unrecorded, further complicating this issue.

Regardless of which vaccines have been made available, an increase in the number of SIAs conducted in a health zone was associated with greater overall rates of seroprevalence and also with higher markers of knowledge and outreach activities, such as having someone visit your home to discuss polio or hearing about a vaccination campaign in your area. The association between the number of SIAs conducted in a health zone and seroprevalence rates remained even after controlling for other key factors such as age, sex, and parental knowledge factors. In these rural and semi-rural communities, the dissemination of medical resources and information has been an ongoing challenge. One key concern has been that official reports of the number of villages reached and vaccine units distributed may not reflect the realities of the fieldwork of a vaccine campaign. A positive relationship between the number of campaigns conducted, seroprevalence, and outreach provides useful evidence that SIAs are indeed reaching their target communities and impacting vaccine coverage rates.

While SIAs may be doing their part to improve poliovirus competencies among adults in this region, knowledge of poliovirus and its mechanisms of transmission is still lacking, leaving barriers to community-based prevention and control. Overall, 60.7% of parental guardians in this survey knew of polio disease, but just 25.6% understood how it is transmitted. As paralysis occurs in just 1% of cases [17], there is a pressing need to expand communal understanding of polio’s more common symptoms, like fever and headache, as potential indicators of infection. In the context of tropical sub-Saharan Africa, this can be a challenge as both symptoms are nonspecific and can indicate infection with a number of other endemic agents, including malaria. However, in a region such as DRC with a history of frequent cVDPV2 cases, community members should be educated to identify persistent fever and diarrhea as potential signs of poliovirus infection. There were some inconsistencies in parental knowledge of polio. For example, 132 (13.7%) participants identified paralysis as a symptom of polio but responded that they did not know what polio disease was. This could be indicative of different frameworks for disease models that can exist between communities and health practitioners or a reflection of how polio has been communicated to families by health authorities as a cause of paralysis rather than an enteric disease. While surveys were administered by local interviewers in local languages, perhaps future surveys should present these topics in a way more reflective of local conceptions of pathogens and disease. This study was limited by a few important factors. One key limitation is that the laboratory methods used in this study were not able to distinguish between the presence of neutralizing antibodies due to vaccination or natural infection. This was further obscured as several SIAs were launched in response to cVDPV2 cases. Consequently, seroprevalence rates cannot be fully interpreted as a reflection of vaccine coverage. Doing so would likely inflate vaccination rates, particularly for the rates of type 2 since cVDPV2 was circulating in the time leading up to specimen collection. To combat this, we collected vaccine information from vaccine cards, which record all vaccinations a child has received [18]. However, only 13.1% of participants had a vaccine card, preventing our ability to perform an unbiased sub-analysis. Other possible limitations include those arising from sampling bias. Yet, as knowledge and outreach factors were also positively associated with the number of SIAs conducted in a health zone, we can conclude that the strength of association between SIAs and seroprevalence is likely not an artifact of reverse causation. 

To combat VDPV2s, a novel OPV2 (nOPV2) vaccine has been developed that is more genetically stable than the current mOPV2. DRC was a target country for the initial rollout of nOPV2 in March 2021. Since then, VDPV2 outbreaks in the county have waned compared with the years prior [8]. However, despite nOPV2′s successes, there has been evidence that cVDPV2 have been seeded from this new vaccine [19]. While Sabin-strain vaccines have been highly effective in polio eradication since the 1960s, endgame polio vaccination strategies must pay careful attention to the risk of wild-type reversion when using oral vaccines. In low-resource settings, improving health infrastructure for IPV, an injectable vaccine, should be considered equally as important. 

## 5. Conclusions

Overall, this survey provides an update to the 2014 and 2016 polio serosurvey conducted in the southeastern Katanga region of the DRC. Since then, polio seropositivity rates among children under 5 years of age have fallen to 38–44%. As this region has experienced multiple cVDPV2 outbreaks since 2011 and is a key area for cVDPV2 eradication, a thorough and widespread vaccination strategy is of paramount importance. 

## Figures and Tables

**Figure 1 vaccines-12-00246-f001:**
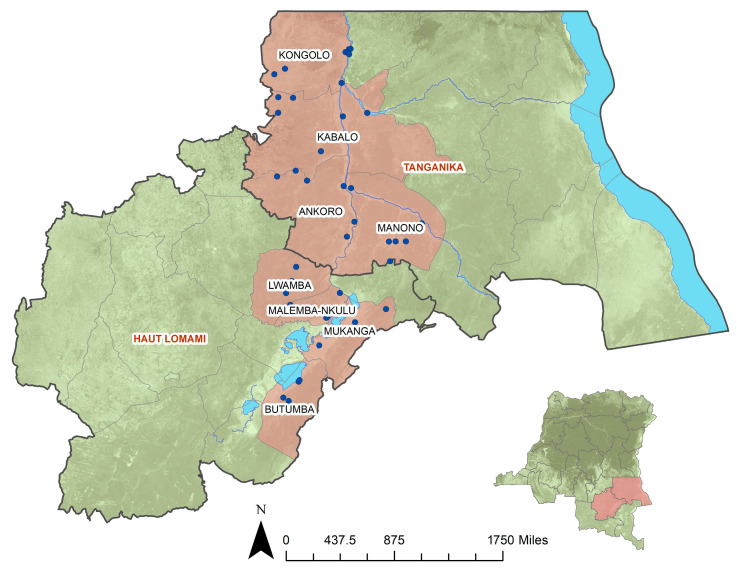
Map of Haut Lomami and Tanganyika provinces and selected study site locations.

**Figure 2 vaccines-12-00246-f002:**
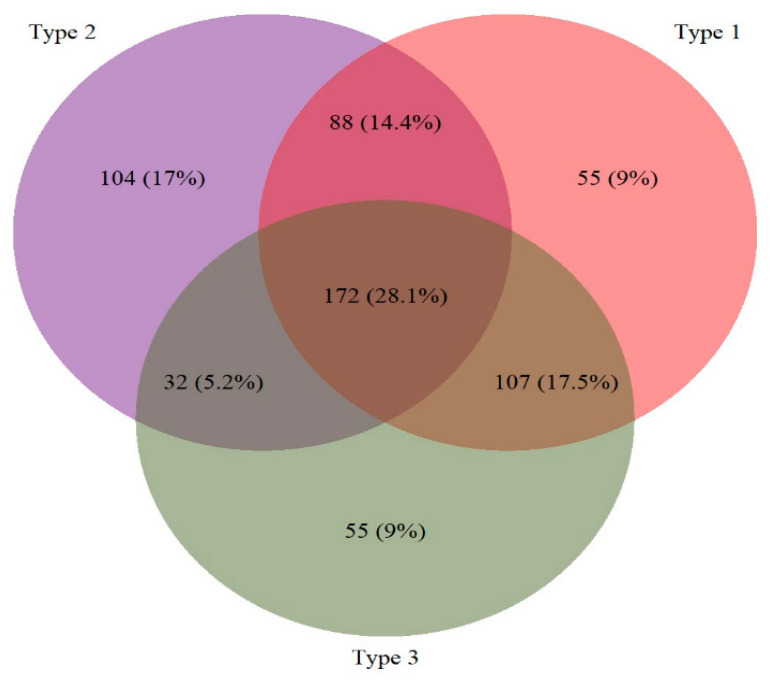
Prevalence of serologic profiles among participants with seroprotective titers (N = 613).

**Figure 3 vaccines-12-00246-f003:**
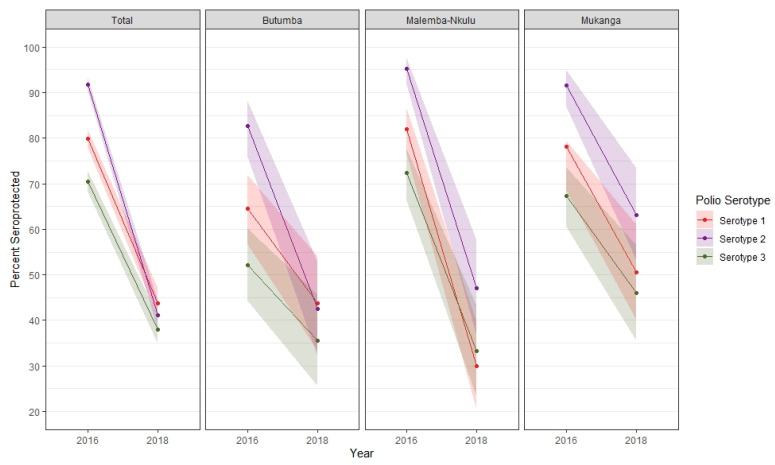
Polio seroprevalence estimates from 2016 to 2018.

**Figure 4 vaccines-12-00246-f004:**
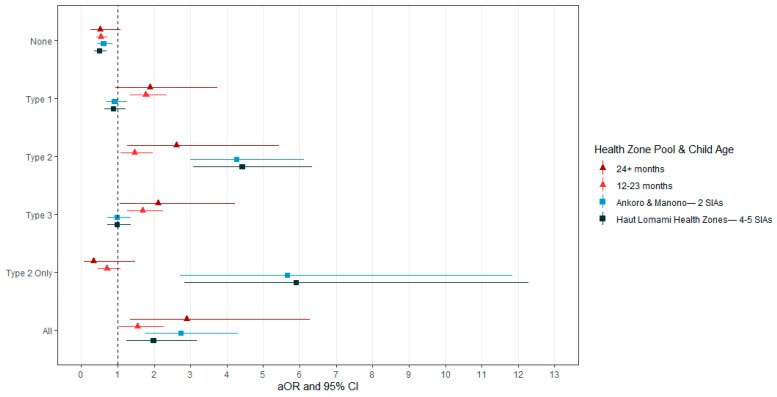
Odds of seroreactivity by health zone pool and age group. Model used is estimating odds of seroreactivity by health zone history adjusted by age and sex (non-significant and not shown). Reference health zone: Kabalo and Kongolo, which had no cVDPV2 cases and 0 SIAs. Reference age: 6–11 months old.

**Table 1 vaccines-12-00246-t001:** Parental or guardian knowledge of polio disease (n = 964).

	n	%
Do you know what polio is?	585	60.68
Do you know how polio is spread?	247	25.62
If your child was to get sick with polio, what symptoms could they get?		
Paralysis	717	74.38
Fever	370	38.38
Diarrhea	314	32.57
Don’t know	141	14.63
What would you do if your child suddenly was unable to walk?		
Take them to a local health care practitioner	452	46.89
Take them to a doctor	245	25.41
Take them to a hospital	133	13.8
I don’t know	61	6.33
Do nothing or wait	41	4.25
Treat at home with over-the-counter medicines	29	3.01
Has someone ever come to your home to give you information about polio?	765	79.36
In the last year have you heard about any polio vaccine campaigns?	853	88.49
In what ways have received campaign information about polio?		
Community/Village health volunteer	182	18.88
TV commercial	890	92.32
Radio Commercial	948	98.34
Poster	247	25.62

## Data Availability

Data may be made available upon request.

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
