# Peer review of "Poliovirus-Neutralizing Antibody Seroprevalence and Vaccine Habits in a Vaccine-Derived Poliovirus Outbreak Region in the Democratic Republic of Congo in 2018: The Impact on the Global Eradication Initiative"

_vaccines, 2024, doi:10.3390/vaccines12030246_

Round 1

Reviewer 1 Report

Comments and Suggestions for Authors

Reviewer’s report

Manuscript:

Poliovirus Seroprevalence and Vaccine Habits in a Vaccine-Derived Poliovirus Outbreak Region in the Democratic Republic of Congo, 2018

In this work, the authors have evaluated the immunization rate against poliovirus types 1, 2 and 3 in an outbreak region in Democratic Republic of Congo in 2018 and tried to explain the high rate of cVDPVs reported cases vs under immunization against poliovirus

This study focuses on the southeastern region of the Democratic Republic of the Congo (DRC), which experienced outbreaks of circulating vaccine-derived poliovirus type 2 (cVDPV2) following the global switch from trivalent oral polio vaccine (tOPV) to bivalent oral polio vaccine (bOPV) in 2016. The study aimed to update population seroprevalence data for markers of poliovirus immunity, understand risk factors for under-immunization, and explore parental knowledge and behaviors influencing vaccine decision-making.

A cross-sectional population-based survey was conducted in March 2018 in eight health zones across Haut Lomami and Tanganyika provinces.

The study reveals a declining trend in seroprevalence in the region since previous surveys in 2014 and 2016. Challenges in vaccine distribution, uptake, and availability contribute to diverse serologic immunity profiles. SIAs play a crucial role in improving overall seroprevalence, but knowledge gaps persist, necessitating enhanced community education efforts. The study is limited by the inability to distinguish between vaccine-induced and naturally acquired antibodies. Sampling bias and limited vaccine card availability further impact the interpretation of results.

This research underscores the importance of targeted vaccination campaigns and comprehensive community education in regions prone to polio outbreaks. The findings contribute to ongoing efforts to understand and address barriers to achieving and sustaining polio immunity.

Some comments and revisions should be incorporated into the manuscript to enhance clarity for the readers:

1.     Title: Consider rephrasing the title adding the importance of the study in the last phase of poliovirus eradication

This is a proposition:Seroprevalence of Poliovirus Among Children in a Region affected by a Vaccine-Derived Poliovirus Outbreak in the Democratic Republic of Congo, 2018: Impact on the Global Eradication Initiative.”

2.     Keywords: Please add cVDPV, SIAs, OPV, IPV, nOPV2

3.     Tables and figures:

a.     Consider placing Table 1 and Table 2 in the supplementary material to lighten the content of the main text.

b.     Table 1 lacks clarity and is challenging to comprehend.

c.     Provide more detailed and descriptive titles for all figures and tables to enhance clarity.

4.     Results:

a.     In the results section, the authors initially presented the percentages of protective antibodies in 964 participants (lines 152-153). However, a discrepancy arises in lines 155-156, where it is stated that n=351 had no antibodies. Consequently, Figure 2 was apparently based on n=653 participants. This creates confusion as it is essential to highlight the significant percentage of participants without protection against poliovirus. Additionally, the initial percentages presented are not accurate, and the correct percentages are reflected in Figure 2( calculation on the 653 with antibodies). Therefore, the entire paragraph requires rephrasing to ensure clarity and accuracy in conveying the information.

The accurate rates should be also added in the abstract section

b.     Participants' ages ranged from 6 to 59 months; however, in Table 1, only three age groups up to 35 months are represented, what about the age group 35-59 months?

c.     Please revise all the percentages in the table 1

5.     Discussion: Lines 241-242:

a.     the sentence “Additionally, 10.8% of our study (n=104) was seroreactive to only type 2” could be rephrased to “Moreover, a seroreactivity to only type 2 was observed in 10.8% of our study participants (n=104)”.

b.     In the discussion it is important to introduce the impact of your study on the global eradication initiative, what are your prevision while introducing the new nOPV2

c.     Your perspectives in continuing the seroprevalence evaluation during the Five last years

6.     References:

a.     Some references are outdated, and it would be beneficial to update them. For instance, in the background section, the authors make reference to current information but cite publications from 2016.

b.     Add some updated references to enrich the study especially regarding the nOPV2 and the current situation

Author Response

thank you very much for the thoughtful comments. Attached are my responses

Reviewer 2 Report

Comments and Suggestions for Authors

The manuscript "Poliovirus Seroprevalence and Vaccine Habits in a Vaccine-Derived Poliovirus Outbreak Region in the Democratic Republic of Congo, 2018" shows the seroprevalence of the protective antibodies against poliovirus types 1-3 and discuss the effect of the vaccination rate on the emerging vaccine derived poliovirus 2 in poorly vaccinated areas of the country.

This is an important topic and concerns not only one country but also poses a threat to the health security in the world since the virus has been eradicated and the only remaining type has only reported from a very limited area. The importance of the issue is the modeling of the data to show how a vaccine effort can or may not be successful.

There are issues need to be dealt with in the manuscript.

The introduction is limited to the vaccination history of the poliovirus and the outbreaks of the virus in recent years but has almost nothing to introduce about the vaccination policies or vaccine coverage in the DRC. Adding a few lines might help to see the whole picture better in this case.

It is good to talk about the vaccines used and how the prevalence of the antibodies for various types especially types 1 and 3 are so much different. You have mentioned the immunogenicity issue with the PV3 but the titer trends were not shown. Please add the mean or median of titers to each virus type as well.

The study also seems not to have a power calculation to justify the results of the survey being representative of the country.

The data in table 1 is not introduced properly. Ther are three numbers in the cells of which one id Col%. I cannot work out what they mean. Please clarify.

You have reported that about 89% knew there is a poliovirus campaign but only 60 % knew what polio was. How this can be explained? Please discuss.

You have stated that in Kabalo and Kongolo health zones, 46.1% of the sample population had no poliovirus neutralizing antibodies yet had no cVDPV2 cases. How this can be explained?

In figure 4 the odds of having no antibody was less than 1 in younger ages compared to the refence. Do you know if there are maternal antibodies in newborns in the region? How do you explain this? Please discuss.

Comments on the Quality of English Language

Seems to be alright

Author Response

(The authors gave the same response as above.)

Reviewer 3 Report

Comments and Suggestions for Authors

This is a well written study that serves to extend our understanding of the population dynamics of polio vaccination in the DRC and likely elsewhere.  I recommend being more accurate when discussing the data including in the title as the manuscript is not about 'poliovirus' seroprevalence but poliovirus neutralizing antibody seroprevalence

While the data presented is comprehensive, there are some important technical considerations about the comparative analysis of vaccine efficacy (seroprevalence) over time that need to be clarified.

The addition of simple Figure legends and some Table footnotes would improve this issue as well as overall readability of the manuscript.

Table 1 – the data presented in the columns is not easy to follow.  For the population in most of the column sections there is an upper and lower n, the latter being unclear, with the Col % lined up with the latter.  This needs to be formatted more clearly with the upper and lower ‘n’s described.

The data in Figure 3 are problematic in comparing historical data with the more recent study data.  Were the tests used performed with similar samples using the same technology at the same institute?  Were the ages of the tested individuals and time after vaccination comparable?  This information should all be provided in a figure legend.  This is a key piece of study data that needs to be explained in more detail technically so that the reader can better judge whether the difference is a technical artefact, differences in the age groups assessed, due to increased time after vaccination, or some other unanticipated group characteristic.  The time between vaccination and testing is an important parameter that appears to be absent from the data collected and together with differences in immunological maturity at vaccination may somewhat explain differences in seroprevalence between age groups.

Typo line 46 - 'remaning'

Author Response

(The authors gave the same response as above.)

Round 2

Reviewer 1 Report

Comments and Suggestions for Authors

The manuscript is now more comprehensible and it is now more comprehensible. It can be published